# Development and Comprehensive Characteristics of Thermosensitive Liquid Suppositories of Metoprolol Based on Poly(lactide-*co*-glycolide) Nanoparticles

**DOI:** 10.3390/ijms232213743

**Published:** 2022-11-08

**Authors:** Maria Bialik, Joanna Proc, Anna Zgadzaj, Karolina Mulas, Marzena Kuras, Marcin Sobczak, Ewa Oledzka

**Affiliations:** 1Department of Analytical Chemistry and Biomaterials, Faculty of Pharmacy, Medical University of Warsaw, 1 Banacha Str., 02-097 Warsaw, Poland; 2Department of Environmental Health Sciences, Faculty of Pharmacy, Medical University of Warsaw, 1 Banacha Str., 02-097 Warsaw, Poland

**Keywords:** metoprolol tartrate, thermosensitive liquid suppository, nanoparticles, lactide and glyccolide copolymers, antihypertensive drug, hypertension, drug delivery system

## Abstract

Thermosensitive liquid suppositories (LSs) carrying the model antihypertensive drug metoprolol tartrate (MT) were developed and evaluated. The fundamental purpose of this work was to produce, for the first time, liquid MT suppositories based on biodegradable nanoparticles and optimize their rheological and mechanical properties for prospective rectal administration. The nanoparticle system was based on a biodegradable copolymer synthesized by ring opening polymerization (ROP) of glycolide (GL) and L,L-lactide (LLA). Biodegradable nanoparticles loaded with the model drug were produced by the o/o method at the first stage of the investigation. Depending on the concentration of the drug in the sample, from 66 to 91% of MT was released over 12 h, according to first-order kinetics. Then, thermosensitive LSs with MT-loaded biodegradable nanoparticles were obtained by a cold method and their mechanical and rheological properties were evaluated. To adjust the thermogelling and mucoadhesive properties for rectal administration, the amounts of major formulation components such as poloxamers (P407, P188), Tween 80, hydroxypropylcellulose (HPC), polyvinylpyrrolidone (PVP), and sodium alginate were optimized. The in vitro release results revealed that more than 80% of the MT was released after 12 h, following also first-order kinetics. It was discovered that the diffusion process was dominant. The drug release profile was mainly governed by the rheological and mechanical properties of the developed formulation. Such a novel, thermosensitive formulation might be an effective alternative to hypertension treatment, particularly for unconscious patients, patients with mental illnesses, geriatric patients, and children.

## 1. Introduction

Metoprolol tartrate (MT) (Figure 1) is a popular selective β_1_-blocker used in the treatment of hypertension and cardiac disorders such as angina pectoris, arrhythmias, and myocardial infarction. MT is almost completely absorbed after oral administration; its absorption rate exceeds 90% and it undergoes significant hepatic first-pass effect elimination, with a reported systemic bioavailability of 40% and a half-life of 3–4 h. Despite significant individual differences, around 5% of orally administered MT is eliminated unchanged in the urine [1].

Although oral administration is the simplest, most convenient, and preferred method for many therapeutic agents, it is not the ideal route for all pharmaceuticals. As stated above, certain therapeutic compounds, such as MT, are restricted due to constraints such as low bioavailability and extensive hepatic first-pass metabolism [2]. To address these issues, researchers attempted to develop novel formulations based on biodegradable macromolecular prodrugs, self-microemulsion drug delivery systems (SMEDDS), or nano-drug delivery systems (NDDSs), with the goal of improving drug bioavailability, therapeutic profile, and plasma drug concentration. Unfortunately, the oral bioavailability of a lipophilic drug is still a major issue. As a result, scientists are constantly challenged to create more innovative drug delivery systems (DDSs) [3].

Rectal administration may be a feasible alternative to oral administration for effectively addressing bioavailability problems and increasing drug therapeutic potential. Furthermore, among the several noninvasive ways accessible, the rectal route is a safe alternative way to administer drugs, particularly when the oral route is not feasible due to patients’ problems with swallowing, nausea, and vomiting, and for infants, children, and geriatric patients [2,3].

Conventional solid suppositories often cause patient discomfort and a sense of alienation, resulting in patient refusal and low compliance. Furthermore, a non-mucoadhesive solid suppository may reach the end of the colon, causing the drugs carried to undergo first-pass effect. Furthermore, because to its fast melting in the rectum, it has no sustained drug-release function. The ideal suppository should be simple to apply and stay at the administration site to avoid the first-pass effect in the liver [4,5,6,7].

To resolve these concerns, the development of thermosensitive liquid suppositories (LSs) is desirable. This innovative formulation is liquid at ambient temperature and gel at body temperature. As a result, it reduces patient discomfort during administration. Furthermore, it has sufficient gel strength so that it does not leak out of the anus. The thermosensitive LSs have adequate bio-adhesive force and do not reach the end of the colon, preventing the first-pass effect in the liver [2,8,9]. The thermosensitive LSs have already been developed as vehicles for a number of drugs, including: acetaminophen [10], epirubicin [11], ibuprofen [12], etodolac [7], metoclopramide [13], carbamazepine [14], and insulin [15,16]. However, only three gelling rectal dosage forms for antihypertensive drugs have been developed thus far, namely for propranolol [17], diltiazem [18], and candesartan [4].

Mucoadhesive liquid suppositories loaded with propranolol were produced by adding mucoadhesive polymers (hydroxypropyl cellulose, polyvinylpyrrolidone, Carbopol, polycarbophil) to formulations of thermally gelling suppositories containing poloxamer 407 (15%) and poloxamer 188 (15%). The suppositories’ properties differed depending on the amount of mucoadhesive polymer added. Propranolol was released from such formulations according to first-order kinetics, with rectal bioavailability ranging from 61% to 85%. The authors concluded that retaining in the rectum drug via the addition of appropriate mucoadhesive polymers appears to be a critical factor in avoiding first-pass hepatic elimination and thereby increasing the drug’s bioavailability [17]. Similarly, mucoadhesive polymers (Carbopol 974P, hydroxypropyl methylcellulose, poly-vinylpyrrolidone, polycarbophil) were added to a formulation containing poloxamer 407 (20%) and poloxamer 188 (10%) to create thermosensitive mucoadhesive liquid suppositories with diltiazem. The addition of these polymers increased gel strength and mucoadhesive force while slowing drug release. Diltiazem hydrochloride formulated as a thermosensitive liquid suppository for rectal administration has the potential to avoid first-pass effect and increased bioavailability [18]. The last example of using an antihypertensive drug in the formulation of a thermosensitive liquid suppository is chitosan-based microspheres with candesartan. First, chitosan-based microspheres were formed using a single emulsification technique, and then liquid suppositories were developed using a cold preparation method. The prepared microspheres were combined into the poloxamer 407 (18%) gel. Chitosan demonstrated good mucoadhesion properties in the rectum, retarded drug release by up to 12 h. The authors concluded that chitosan microspheres incorporated into liquid suppositories are a viable alternative to other conventional suppositories [4].

In recent years, there has been a surge of interest in biocompatible and biodegradable polymers (also known as “biomedical polymers”). They can be utilized, among other things, in the technology of sustained dosage forms, which allows drugs to be dosed with the required pharmacokinetics, enhance their solubility in water or lipids, durability, and absorption. In general, the sustained release dosage form is designed to maintain a constant drug therapeutic concentration in the plasma, thereby decreasing the number of daily doses. It may also minimize the risk of adverse effects and toxicity. Furthermore, this solution has the potential to increase drug pharmacokinetics and bioavailability [19]. Given the above biodegradable copolymer poly(lactide-*co*-glycolide) (PLGA) was used as vehicle for MT in this study.

As a result of the foregoing, in the present study biodegradable copolymer PLGA was successfully synthesized via a ring opening polymerization of L,L-lactide and glycolide initiated by poly(glycol ethylene) (PEG). The fully characterized copolymeric carrier was then used to produce MT nanoparticles by the oil-in-oil (o/o) method. The particle size, zeta-potential, polydispersity index, and morphology of the developed material were all determined. Following that, the in vitro drug release study from the obtained nanoparticles was investigated. It was discovered that the drug release profile was primarily determined by the concentration of MT in the sample. Various thermosensitive liquid suppositories were prepared in the final step of this investigation, using varying amounts of mucoadhesive polymers and MT-loaded nanoparticles. The gelation temperature, gelation time, gel strength, and viscosity of the formulation were all tested. Two thermosensitive LSs with optimum mechanical and rheological properties were chosen for in vitro MT release studies. It was discovered that thermosensitive LSs properties governed the MT release rate.

## 2. Results and Discussion

### 2.1. Synthesis and Characterization of the Biodegradable PLGA Carrier

In the first stage of the research, LLA and GA copolymer (PLGA) was synthesized as a biodegradable MT carrier. Ring-opening polymerization (ROP) of LLA and GL was carried out with Zn(Et)_2_ as a catalyst, and PEG was employed as a biosafe co-initiator of the ROP process. ROP was carried out in bulk at 125 °C (Table 1).

The structure of the PLGA carrier was investigated using ^1^H or ^13^C NMR (Appendix A (^1^H NMR spectra of PEG-PLGA) and Appendix A (^13^C NMR spectra of PLGA, Appendix A). The specific signals observed on the spectra proved the structure of the resulting material. The major signals related to methylene or methine proton signals terminated by hydroxyl end groups of PLGA chains are plainly seen in the Appendix A (^1^H NMR spectra of PEG-PLGA, Appendix A). The ^1^H NMR study revealed that the initiator (PEG) used efficiently initiated ROP of LLA and GL. The FT-IR spectra provide additional evidence of the successful synthesis of PLGA (Appendix A, FT-IR spectra of PEG-PLGA, Appendix A).

The ^13^C NMR spectra of PLGA produced in the presence of PEG initiator and Zn(Et)_2_ catalyst show five highly resolved signals: LLGG (169.68 ppm), LLLL (169.64 ppm), GLG (169.58 ppm), GGGG (166.52 ppm), and GGLL (166.47 ppm), where L denotes the lactyl unit -CH(CH)-CO-O-, and G denotes the glycolyl unit -CH,-CO-O- (Figure 2). The signal GLG, depicted in Figure 2, can only be formed as a result of type II intermolecular transesterification (Table 2).

In order to fully characterize the microstructure of the synthesized carrier, ^1^H NMR spectra was also engaged. The well-resolved region of methylene protons of the glycolide comonomer was selected for this purpose (Figure 3).

The average lengths of lactidyl and glycolidyl blocks, in addition to the second mode of transesterification, were determined using the formulae provided by Kasperczyk et al. [20]. The average lengths of lactidyl and glycolidyl blocks were 1.04 and 1.55, respectively. The second mode of transesterification (T_II_) was 12.3 (Table 1). Based on the calculated results, we can deduce that the resulting copolymer PLGA had a predominantly atactic structure.

The *M*_n_ and dispersity (*Đ*) of the obtained PLGA copolymer were determined by SEC-MALLS technique (Table 1). The results collected in Table 1 show that the synthesized copolymer has a rather narrow *Đ* index and the *M*_n_ value of 14700 g/mol.

The residual Zn content from the catalytic system was removed by washing the synthesized biodegradable carrier with dilute hydrochloric acid. The Atomic Absorption Spectrometry (AAS) technique was used to evaluate the presence of metallic residues, which revealed a value of 0.368 ppm, i.e., 0.0000368%. According to European Pharmacopoeia standards, the zinc level in materials in contact with blood or blood components cannot exceed 0.2% [21]. As a consequence, the achieved result is satisfactory, and the synthesized carrier is suitable for biomedical applications.

Since it is widely known that materials utilized in the medical and pharmaceutical areas must meet certain criteria, the synthesized copolymeric matrix was also subjected to cyto- and genotoxicity assays. The data in Table 2 and Table 3 show that the synthesized carrier is not cytotoxic in the neutral red uptake assay (Table 2) and is not toxic for *Salmonella typhimurium* TA1535 (G > 0.5) with and without metabolic activation. Furthermore, it did not inhibit genotoxic activity (IR < 1.5) (Table 3).

### 2.2. Synthesis and Characterization of MT-Loaded Nanoparticles

In the following stage of this investigation, synthesized and characterized biodegradable carrierwas used to develop nanoparticulate system loaded with antihypertensive drug—MT. Three different nanoparticulate samples with varying amounts of MT were produced (O1, O2, O3) (Table 4).

The DLS analysis revealed that the produced nanoparticles had average sizes of 225 nm (O1), 269 nm (O2), and 235 nm (O3) (Table 4). The dispersity values were in the 0.005–0.14 range, indicating a narrow particle size distribution. Zeta potential measurements revealed that all nanoparticles have a negative surface charge (from 18.53 to 20.95 mV), which is most likely explained by the presence of the hydroxyl end-group in the carrier chain [22]. Additionally, Appendix A (The dependence of the nanoparticles size on the intensity for O2 sample, Appendix A) depicts the relationship between nanoparticle size and intensity for an O2 sample. The TEM technique was also employed to confirm the DLS results (Figure 4).

The DLS analysis results are supported by the TEM method. As was shown in Figure 4, the size range of nanoparticles samples was 250–300 nm. Furthermore, as was indicated in Table 4, the EE (encapsulation efficiency) value was in the range of 33–43%. The highest EE value was found for the O3 sample (43%), while the lowest was for O1 sample (33%). These differences are most likely related to the different amounts of MT used for nanoparticles preparation.

### 2.3. MT Release Studies from the Prepared Nanoparticles

The drug release kinetic was evaluated to estimate the drug release characteristics in the developed materials. MT was employed as the model hydrophilic antihypertensive drug. The drug release profiles of 3 different nanoparticle samples (O1, O2, O3) were examined as a relationship between cumulative drug release and time (Figure 5). As we can easily observed on the presented Figure 5, the release rate reached a plateau after 10 h for all studied samples. After 12 h of incubation, the O3 sample released the highest amount of MT (91%), while the O1 sample released just 67%. The observed differences should be accounted by a different MT concentration in the samples (Table 4). However, there were no significant differences in the active substance release profiles from the MT-loaded nanoparticle samples.

The collected data were fitted to mathematical models, especially the zero-order, first-order, Higuchi, and Korsmeyer-Peppas, to investigate the drug release profiles. The findings are presented in Table 5. Given the slight increase in the cumulative release during the plateau phase, kinetic calculations were performed using data from the first 10 h of the experiment.

The examination of R^2^ values for zero-order and first-order kinetics revealed that nanoparticle samples, O2 and O3, released MT in a pattern similar to that of first-order kinetics. Furthermore, the drug release profiles followed the Korsmeyer-Peppas model closely; R^2^ for O1, O2, O3 samples was 0.953, 0.928, and 0.956, respectively. The release exponent (n) was 0.399 (O1) and 0.301 (O3), showing that the drug was released primarily by diffusion (n ≈ 0.45) (the release exponent for O2 sample equals 0.934) [23]. The dominance of the diffusion mechanism and first-order kinetics is supported by the matrix’s relatively long degradation rate and the hydrophilic nature of MT. According to the data, we can conclude that MT rate of release is faster than the nanoparticle degradation rate. The pathway through which the solvent penetrates the pores of the insoluble matrix and the route through which the MT is eluted, increases longer as time passes, and the concentration gradient decreases. Because of this phenomena, the drug release rate decreases over time, resulting in first-order kinetics. The well-fitting of the release data to the Higuchi model for samples O1 and O3 verified the above results; the R^2^ values for O1 and O2 were 0.905 and 0.925, respectively. The exponent value n for sample O2 was 0.834 according to the Korsmeyer-Peppas model for the diffusion-degradation-controlled release mechanism. According to the literature, when n is close to 0.45, the process is dominated by Fickian diffusion. When n is close to 0.89, the drug release is controlled by degradation [23,24]. The fitted plots have been added to Appendix A as examples (Appendix A (the fitted plot for sample O—zero-order kinetic), Appendix A (the fitted plot for sample O1—first-order kinetic, Appendix A (the fitted plot for sample O1—Korsmeyer-Peppas model and Appendix A (the fitted plot for sample O1—Higuchi model).

The nanoparticle sample O3 was selected for further investigation since it had the highest EE value and the highest amount of MT released.

### 2.4. Synthesis and Characterization of Thermosensitive LSs Loaded with MT Based Biodegradable Nanoparticles

The poloxamer-based thermosensitive LSs were developed primarily to improve MT bioavailability by rectal administration. Rectal delivery would effectively mitigate the disadvantages of the oral and intravenous methods. Table 6 shows the various thermosensitive LSs compositions created from varying concentrations of P188, P407, MT, MT loaded biodegradable nanoparticles (O3), Tween 80, HPC, sodium alginate, and PVP. The purpose of this study was to assess the impact of various concentrations of the formulation components on the rheological and mechanical properties of thermosensitive LSs. As a result of the foregoing, the gelation temperature, gelation time, viscosity, and gel strength of the developed novel formulation were determined.

The gelation temperature is the temperature at which the liquid phase transforms to the gel phase. The gelation temperature range for rectal administration is 30–37 °C [2]. The results from Table 7 indicate that the gelation temperature decreased as the concentration of P407 increased. In most cases, the addition of MT and MT loaded nanoparticles lowered the gelation temperature value. The thermosensitive LSs with PVP components had the highest average gelation temperature of 42 °C, while the suppositories with HPC components had the lowest (36.8 °C). Nevertheless, for the following steps of the investigation, we used thermosensitive LSs samples containing MT or MT-loaded nanoparticles with acceptable gelation temperatures (particularly, samples 12, 13, 17, 19, 20, 21, Table 6).

During thermosensitive LSs preparation, the gel strength parameter is critical in determining the conditions that allow for simple insertion of the formulation and no anus leak. The thermosensitive LSs leak from the anus at low gel strength value, but LSs with high gel strength are difficult to insert. The thermosensitive LSs with the ideal gel strength value of 10–50 s will remain in the upper region of the rectum and will not leak out from the anus [2,14]. In relation to the above-mentioned, Table 8 displays the gel strength values for the developed thermosensitive LSs with the optimal gelation time parameter. Each of the tested thermosensitive LSs had an adequate gel strength value.

The optimal viscosity for the thermosensitive LSs should equal min. 4000 mPa·s at 36.5 °C. Simultaneously, the gelation period varies depending on the suppository composition, but is typically 2–8 min [2]. Table 8 displays the findings for the selected samples. Only samples 19 and 21 achieved the necessary viscosity at body temperature. Their gelation times were 3 and 4 min, respectively. Because the other thermosensitive LSs did not reach the necessary viscosity, their gelation time values were not determined.

The produced thermosensitive LSs (samples 19 and 21; the visual image of the representative sample can be found in the Appendix A, the visual image of the produced thermosensitive LS)) were further tested using the neutral red uptake assay. None of the tested materials reduced the viability of the BALB/c 3T3 cells below 70 mL as compared to the untreated control. As a result, all tested samples were categorized as non-cytotoxic in the neutral red uptake assay (Table 9). The influence of MT concentration in some samples on the findings was eliminated.

### 2.5. MT Release Kinetics from the Prepared Thermosensitive LSs

Since the O3 sample fitted the first-order kinetics closely and released the highest amount of MT, it was chosen for the preparation of thermosensitive LSs. As a consequence, two thermosensitive LSs with the optimum mechanical and rheological properties (samples 19 and 21) were prepared and utilized in MT release experiments. The thermosensitive LSs based on MT-loaded nanoparticles were transparent and sol-state at room temperature, and they were simple to inject with a 22 Gauge needle. The solutions formed a stable gel after 5 min at 37 °C. The drug kinetics release from the selected thermosensitive LSs was evaluated over a 12-h period at pH 7.0 ± 0.05 and 37 °C (Figure 6). The plot’s ordinate was calculated using the cumulative amount of MT released.

It was observed that the difference in drug release rate found for the thermosensitive LSs was mostly due to their viscosity. The rate of in vitro MT release decreased in direct proportion to the formulation viscosity. After 12 h of incubation, the percentage of MT released was around 80% for the sample 19 (viscosity = 10229.9 mPa·s) and 88% for the sample 21 (viscosity = 4225.71 mPa·s). It was also discovered that MT from the examined thermosensitive LSs was released in a relatively regular and continuous way. Once the MT release profiles were analyzed, it was shown that the drug release rate decreased with time and reached a plateau after 11.5 h.

To evaluate the drug release profiles, the collected data were also fitted to mathematical models, especially the zero-order, first-order, Higuchi, and Korsmeyer-Peppas. Table 10 shows the outcomes.

The findings collected suggest that MT is released in a rather controlled manner from the formulations developed in this research. For the near-first-order kinetics model, determined R^2^ values ranging from 0.893 to 0.977. For the samples 19 and 21, the R^2^ values from the Korsmeyer-Peppas model were 0.878 and 0.926, respectively. It is also worth to note that the controlled MT release rate was achieved with no significant “*burst release*”. This implies that the rate of MT release from the manufactured rectal formulations is a closely controlled process. The n exponents were 0.461 (for the sample 19) and 0.449 (for the sample 21), suggesting that the drug was primarily released via diffusion (n ≈ 0.45) [23].

## 3. Materials and Methods

### 3.1. Materials

Glycolide (1,4-dioxane-2,5-dione, ≥99%, Sigma, Poznan, Poland), L,L-Lactide ((3*S*)-cis-3,6-dimethyl-1,4-dioxane-2,5-dione, 98.0%, Aldrich Co., Poznan, Poland), poly(ethylene glycol) 600 (PEG 600, pure, Fluka, Warsaw, Poland), ZnEt_2_ solution (15% diethylzinc in toluene, Aldrich Co., Poznan, Poland), hydrochloric acid (HCl, ChemPur, Piekary Slaskie, Poland), methanol (pure, 99.9%, ChemPur, Piekary Slaskie, Poland), dichloromethane (DCM, CH_2_CL_2_, ≥99.8%, POCh, Gliwice, Poland), chloroform (pure, 99%, ChemPur, Piekary Slaskie, Poland), metoprolol tartrate (>98.0%, TCI, Tokyo, Japan), phosphate-buffered saline (PBS, pH 7.00 ± 0.05, ChemPur, Piekary Slaskie, Poland), Kalliphor^®^ P 188 (Poloxamer 188, Lutrol^®^ F68, Aldrich Co., Poznan, Poland) Poloxamer 407 (Aldrich Co., Poznan, Poland), PVP (M_w_ 40,000, Aldrich Co., Poznan, Poland), Tween 80 (viscous liquid, Aldrich Co., Poznan, Poland), Sodium alginate (Aldrich Co., Poznan, Poland), HPC (TCI, Tokyo, Japan), Liquid paraffin (ChemPur, Piekary Slaskie, Poland), Trifluoroacetate acid (TFA, ≥98%, Sigma, Poznan, Poland), n-heksan (99%, POCh, Gliwice, Poland), stock solution of Zn(II) (concentration 1000 mg/L, Merck, Darmstadt, Germany), 65% nitric acid (HNO_3_, J.T. Baker, Deventer, Netherlands), dialysis membrane Spectra/Por 3 MWCO 3500 (Spectrum Laboratories, Inc., Gardena, CA, USA) were used as received.

### 3.2. Biodegradable PLGA Carrier

#### 3.2.1. Synthesis Route

The copolymeric materials were synthesized by varying the molar ratios of the initiator (PEG 600) to the monomers: GA or L,L-lactide. For the synthesized polymers, the initiator/monomer feed ratios were 1/10/90 (mol/mol). PEG and monomers were precisely weighed and placed in a 50 mL polymerization tube. The tube was then linked to a Schlenk line, and the exhausting-refilling cycle was performed three times. The catalyst ZnEt_2_ was then added in the next step. The tube was then immersed in an oil bath at 120 °C for 48 h. After a period of time, the polymer products were cooled and dissolved in dry chloroform. The copolymer was then precipitated three times from cold methanol containing 5% hydrochloric acid (HCl). The organic phase was evaporated and dried in a vacuum for 48 h.

#### 3.2.2. Microstructural Analysis

The structure of the obtained PLGA copolymers was elucidated by ^1^H and ^13^C NMR techniques. The spectra were recorded using Varian 300 MHz (Palo Alto, Santa Clara, CA, USA) and Agilent Technologies 400 MHz (Santa Clara, CA, USA) spectrometer. The microstructure of matrices was studied using ^13^C NMR and ^1^H NMR spectra of copolymer PLGA.

The LLA and GA conversions were calculated using the following formulae from the ^1^H NMR spectra of the post-reaction mixture:(1)$$conv\;LLA=\frac{ID}{ID+I\alpha \prime };$$
(2)$$conv\;GA=\frac{ID}{ID+I\alpha \prime }$$
where *I_α_* and *I_D_* correspond to integral intensities of signals from methylene protons adjacent to α carbon atoms of *LLA*/*GA* monomer and *LLA*/*GA* units in PLGA copolymer, respectively [25].

The average molecular weight (*M*_n_) and dispersity index (*Đ*) of the obtained copolymer were measured by SEC-MALLS. *M*_n_ and *Đ* were measured using SEC-MALLS instrument (Wyatt Technology Corporation, Santa Barbara, CA, USA) composed of an 1100 Agilent isocratic pump, autosampler, degasser, thermostatic box for columns, a photometer MALLS DAWN EOS (Wyatt Technology Corporation, Santa Barbara, CA, USA) and differential refractometer Optilab Rex (Wyatt Technology Corporation, Santa Barbara, CA, USA). ASTRA 4.90.07 software (Wyatt Technology Corporation, Santa Barbara, CA, USA) was used for data collecting and processing. Two 2× TSKgel MultiporeHXL columns were used for separation. The samples were injected as a solution in methylene chloride. The volume of the injection loop was 100 mL. Methylene chloride was used as a mobile phase at a flow rate of 0.8 mL·min^−1^.

#### 3.2.3. Spectroscopic Analysis

^1^H NMR (DMSO-d_6_, 400 MHz, δH, ppm): 1.56 (PLA, –CH_3_), 3.62 (PEG, -CH_2_), 4.75 (PGL, –C(O)CH_2_O–), 5.16 (PLA, –C(O)CH(CH_3_)O–) (Appendix A (^1^H NMR spectra of PEG-PLGA, Appendix A).

^13^C NMR (CDCl_3_, 300 MHz, δC, ppm): 169 (C=O), 166 (C=O), 70 (PEG, -CH_2_-), 69 (PEG, -CH_2_)), 66 (LA, -CH-), 64 (GA, -CH_2_-), 16 (LA, -CH_3_) (Appendix A, ^13^C NMR spectra of PEG-PLGA, Appendix A).

FTIR (KBr, cm^−1^): 3509 (υ_O–H_), 2993–2882 (υ_C-H_), 1754 (υ_C=O_), 1459–1385 υ_s_(_C-H_) u_as_ (_C-H_), 1135 υ(_C-O-C_) (Appendix A, FT-IR spectra of PEG-PLGA, Appendix A).

#### 3.2.4. Determination of the Catalyst Residue

The detection of the catalyst residue (zinc ions) was carried out using a Perkin Elmer Analyst 400 flame atomic absorption spectrometer equipped with a hollow cathode lamp and a deuterium background corrector at the respective wavelengths using an air-acetylene flame and the manufacturer’s recommended instrumental parameters.

0.500 g of the dried sample was digested with 6.00 cm^3^ of concentrated HNO_3_ closed polytetrafluoroethylene (PTFE) vessels in a microwave oven (Multiwave 3000, Anton Paar (Perkin Elmer). After digestion the solution was transferred into the 100.0 cm^3^ volumetric flask and filled to the mark with Type I (ISO 3696) deionized water of resistivity > 10 MΩ·cm. Zinc contents were determined directly in respective solutions by FAAS with the air-acetylene flame and hollow cathode lamp as light source.

#### 3.2.5. Biological Assays

The cyto- and genotoxicity experiments were performed on the produced copolymeric products in accordance with pharmacopoeia standards for biomaterials. The *umu*-test and the neutral red uptake test were used to measure the cytotoxicity and genotoxicity of the copolymers obtained.

The neutral red uptake test was performed on the basis of ISO 10993 guideline Annex A [26] with BALB/c 3T3 clone A31 mammalian cell line (mouse embryonic fibroblasts from American Type Culture Collection). The quantitative estimation of viable cells in tested cultures was based on their neutral red uptake in comparison to the results obtained for untreated cells. Dead cells have no ability to accumulate the dye in their lysosomes.

The BALB/c 3T3 cells were seeded in 96-well microplates (15,000 cells/100 µL) in DMEM (Lonza) culture medium (supplemented with 10% of calf bovine serum, 100 IU/mL penicillin and 0.1 mg/mL streptomycin) and incubated for 24 h (5% CO_2_, 37 °C, >90% humidity). At the end of the incubation each well was examined under a microscope to ensure that cells form a confluent monolayer. After that culture medium was replaced by the tested extracts. Extracts were prepared by incubation of tested materials in the cell culture medium (1 mg/mL for polymers; 100 mg/mL for suppositories) with reduced serum concentration (5%) at 37 °C for 24 h with shaking and sterilizes by filtration. Cells were treated with four dilutions of each extract in a twofold dilution series for 24 h (three data points for each one). Subsequently treatment medium was removed. Cells were washed with PBS and treated with the neutral red medium for 2 h. Than the medium was discarded, cells were washed with PBS and treated with desorbing fixative (ethanol and acetic acid water solution). The amount of neutral red accumulated by cells were evaluated colorimetrically at 540 nm. Polyethylene film and latex were used as the reference materials (with no cytotoxicity and highly cytotoxic, respectively). At the same time, the impact of the metoprolol content in some samples on the test results was evaluated. The percentage of viable cells in each well was calculated by comparing its OD540 result with the mean result obtained for untreated cells (incubated in the same conditions with fresh culture medium). Samples were considered cytotoxic if they reduced cell survival below 70% compared to the untreated cells (a baseline cells viability). When the BALB/c 3T3 cells viability was not decreased under 70% in the whole range of tested dilutions of the samples, it was considered as non-cytotoxic in this range of concentrations.

The *umu*-test was performed in 96-well microplates according to the ISO 13829 protocol [27] with and without metabolic activation (S9 liver fraction, Xenometrix). Deionized sterile water was used as a negative control, 2-aminoanthracene and 4-nitroquinoline N-oxide were used as positive controls. All tested materials were incubated in phosphate buffered saline (PBS from Gibco) for 24 h, 37 °C with shaking. Before the assay all extracts were sterilized by filtration. All samples were tested in two fold dilution series (three concentrations, three points of date for each one). Clear PBS treated in the same way as all samples were tested as a solvent control.

### 3.3. MT-Loaded Nanoparticles

#### 3.3.1. Preparation of Nanoparticles Loaded MT by o/o Method

Metoprolol tartrate (MT) was loaded using a non-solvent addition-phase separation method. The non-solvent and solvent were liquid paraffin and DCM, respectively. PLGA was dissolved in 25 mL dichloromethane with magnetic stirring at 500 rpm followed by the addition of MT with continuous stirring. The addition of 50 mL liquid paraffin containing 0.1% Tween 80 as a surfactant caused the co-acervation. To solidify the produced nanoparticles, 10 mL of n-hexane was added and the stirring was maintained for 1 h. Finally, the nanoparticles were washed with n-hexane in triplicate and dried in air for 2 h before being dried in an oven for 12 h. The drug-to-polymer ratios used were: 1:1, 1.5:2, and 1.25:2 (*w*/*w*) for MT to PLGA.

#### 3.3.2. Characterization by Transmission Electron Microscope (TEM)

The transmission electron microscope (TEM) was used to assess the size and shape morphology of the nanoparticles. Nanoparticles that had been synthesized were collected on TEM grids. Electron micrographs were taken using a Morada camera on a JEM 1400 transmission electron microscope at 80 kV (JEOL Co., Tokyo, Japan) in the Laboratory of Electron Microscopy, Nencki Institute of Experimental Biology of the Polish Academy of Sciences, Warsaw, Poland.

#### 3.3.3. Characterization by Dynamic Light Scattering (DLS)

Zetasizer Nano ZS (Malvern, UK) was applied for dynamic light scattering measurements of hydrodynamic diameters and zeta potential. Exitation wavelength of DLS instrument was 633 nm (He-Ne laser, power = 5 W) and measurement angle was 173. Polystyrene disposable cuvettes were used during hydrodynamic diameter determination. Measurements were conducted in 25 °C in aqueous solutions. Samples were briefly sonicated before measurement. Zeta potential was measured using standard dip cell for zeta potential measurements equipped with palladium electrodes. All experiments were conducted with four replications.

#### 3.3.4. Calculation of the Encapsulation Efficiency (EE)

The nanoparticles were dissolved in DCM, and the resultant solution was put to 1 mL of PBS (pH = 7.0 ± 0.05) and stirred for 24 h. The mixture was then centrifuged for 10 min at 10,000 rpm, and the supernatant aqueous phase containing MT was measured using the high-performance liquid chromatography (HPLC) technique. The following equation [28] was used to compute the *EE* value:(3)$$EE=\frac{\mathrm{measured}\;\mathrm{drug}\;\mathrm{content}}{\mathrm{theoretical}\;\mathrm{drug}\;\mathrm{content}}\times 100\%$$

### 3.4. Thermosensitive Liquid Suppositories (LSs) Preparation

#### 3.4.1. Preparation of Thermosensitive LSs with MT

A cold method was used to prepare thermosensitive LSs [10]. Various concentrations of mucoadhesive polymers and MT were thoroughly dispersed in distilled water with continuous agitation at room temperature (Table 6). After that, the solution was cooled to 4 °C. Poloxamers were then gradually added to the solution while stirring continuously. The liquid suppositories were kept at 4 °C until a clear solution was formed.

#### 3.4.2. Preparation of Thermosensitive LSs Based on MT-Loaded Nanoparticles

The same cold method was used to create thermosensitive LSs-based MT-loaded nanoparticles [10]. Various quantities of mucoadhesive polymers and MT-loaded nanoparticles were thoroughly dispersed in distilled water with continuous room-temperature agitation. The solution was then cooled to 4 °C. Poloxamers were then gradually added to the solution while stirring continuously. The thermosensitive LSs were kept at 4 °C till they became transparent.

#### 3.4.3. Characterization of the Prepared Thermosensitive LSs

Gelation temperature measurement

A 10 mL transparent vials containing a magnetic bar and 2 g of thermosensitive LSs were placed in a thermostat water bath and heated at a rate of 1 °C/2 min with continuous stirring at 50 rpm. When the magnetic bar stopped moving due to gelation, the temperature indicated on the thermostat was used to estimate the gelation temperature.

Gel strength measurement

To gel the solution, 50 g of thermosensitive LSs were placed in a 100 mL cylinder and gelled in a thermostat at 36.6 °C. The apparatus (35 g) for testing gel strength was placed on the thermosensitive LSs in the cylinder. The gel strength was determined by the time (in seconds) it took the equipment to sink 5 cm into the gelling dosage form.

Viscosity and gelation time measurements

The test was carried out using an Amatek/Brookfield LV DV-II + Pro viscometer (Middleboro, MA, USA) in a closed chamber optimized for small volumes and temperature control. A PolyScience programmable water bath kept the temperature stable. Throughout the measurement, the temperature within the sample chamber was 37 °C. A CPE-52 cone was utilized for measurements, and a pipette was used to measure 2 mL of the tested liquid into the chamber.

Biological Assays

The neutral red uptake assay was carried out in accordance with ISO 10993 guidelines using the BALB/c 3T3 clone A31 mammalian cell line (mouse embryonic fibroblasts from American Type Culture Collection). The quantitative estimate of viable cells in tested cultures was based on their neutral red uptake compared to untreated cells [26].

### 3.5. In Vitro MT Release Studies

2 g of thermosensitive LSs-loaded-MT and 35 mg of thermosensitive LSs-based MT-loaded nanoparticles were injected into vials. The thermosensitive LSs samples were then thermostated at 37 °C for 5 min to form gels, and nanoparticles were suspended in 1 mL PBS buffer (pH = 7.00 ± 0.05). Following that, the materials were immersed in dialysis membrane (3500 Da), placed in 10.0 mL of preheated PBS buffer (pH = 7.00 ± 0.05), and shaken at 130 rpm and 37 °C. After predefined time intervals, the release medium was withdrawn for further testing and entirely replaced with 10.0 mL of preheated fresh PBS buffer. The obtained samples were stored at 18 °C prior to HPLC analysis. The drug release experiments lasted 12 h.

The HPLC apparatus (Beckman Coulter, Miami, Florida, USA) was equipped with an autosampler (Triathlon 900, Spark Holland B.V., Emmen, Netherlands), pomp (Beckman Coulter System Gold^®^ 125NM Solvent Module, Fullerton, CA, USA) and UV/VIS detector (Beckman Coulter System Gold^®^ 166, Fullerton, CA, USA). The analysis was performed at 274 nm with a C18 column (Nucleodur C18 Gravity 150 × 4.6 mm, 5 um, Macherey-Nagel). Solvents A and B (75:25), H_2_O + 0.05% TFA and acetonitrile (ACN), were used to make the mobile phase. The flow rate was set to 1.0 mL/min, the injection volume was set to 20 µL, and the column temperature was set to 35 °C.

#### Mathematic Models

The kinetics and mechanism of MT release were determined by fitting experimental data to theoretical mathematical models, zero-order and first-order, Higuchi and Korsmeyer–Peppas, using the following equations:

Zero-order:(4)F=kt

First-order:(5)$$logF=logF_{0}-\frac{kt}{2.303}\;$$

Higuchi model:(6)$$F=k\sqrt{t}$$

Korsmeyer-Peppas model:(7)$$\frac{M_{t}}{M_{\infty }}=kt^{n}\left(\frac{M_{t}}{M_{\infty }}<0.6\right)$$
where, *F*—the amount of drug released; *F*_0_—the initial concentration of MT, *t*—the release time increment; *k*—the model constant, *M_t_/M_∞_*—the fraction of MT released during time t; *n*—the exponent in the Korsmeyer–Peppas model [29,30].

## 4. Conclusions

To summarize, thermosensitive LSs based on MT-loaded biodegradable nanoparticles were developed and thoroughly investigated. The biodegradable carrier was PLGA, which was successfully synthesized by ROP with LLA and GL monomers and ZnEt2 as a catalyst. The high monomer conversion rate and appropriate product yield demonstrated that the zinc catalyst is adequate in the ROP process. Furthermore, 1H NMR results show that the initiator successfully initiated ROP of LLA and GL monomers. The copolymer produced has a narrow *Đ* index and a Mn of 14700 g/mol. The calculated TII value of the obtained copolymer was 12.3, indicating its atactic microstructure. According to the results of the biological tests, the synthesized carrier is not cytotoxic in the neutral red uptake assay and is not toxic to *Salmonella typhimurium* TA1535 (G > 0.5) with or without metabolic activation. Furthermore, since PLGA has a low zinc content, it is acceptable for biomedical applications.

The o/o method was used to develop MT-loaded nanoparticles with sizes ranging from 225 to 269 nm (DLS technique). It is worth noting, however, that the results obtained were repeatable, and the size of the nanomaterials was confirmed by TEM analysis. Due to the difficulty of producing nanoparticle systems loaded with hydrophilic drugs, our findings demonstrated that the o/o method may be used as an efficient route for producing nanoparticles loaded with hydrophilic MT. The investigated model drug was released by the first-order kinetic model, which corresponded to in vitro drug release data. Depending on the MT concentration in the sample, from 66 to 91% MT was released after 12 h. Furthermore, the calculated release exponent (n) indicates that the drug was released primarily through diffusion.

Finally, thermosensitive LSs were developed based on biodegradable nanoparticles capable of releasing an antihypertensive drug. This innovative formulation was produced with materials that possess the optimal rheological and mechanical properties. Following first-order kinetics, the in vitro release results revealed that more than 80% of the drug was released after 12 h. The obtained data were fitted to the Higuchi and Korsmeyer-Peppas mathematical models, and the diffusion process was discovered to be dominant.

In our opinion, the developed thermosensitive LSs appear promising as short-term DDSs, with potential applications in hypertension therapy, owing to the repeatable and well-controlled drug release rate, as well as biodegradability and biosafety. Furthermore, we would like to emphasize that the formulation produced in our study could be an effective alternative to hypertension treatment, particularly for unconscious patients, patients with mental illnesses, geriatric patients, and children. As was clearly stated, MT has limited application due to its short half-life and low bioavailability. However, we believe that administering MT in the form of thermosensitive LSs can improve its pharmacokinetics and increase its efficiency. It is worth noting at this point that one significant innovation in our project is the use of biodegradable, synthetic polymers to develop thermosensitive LSs. So far, no comparable solution for antihypertensive drugs has been described.

## Figures and Tables

**Figure 1 ijms-23-13743-f001:**
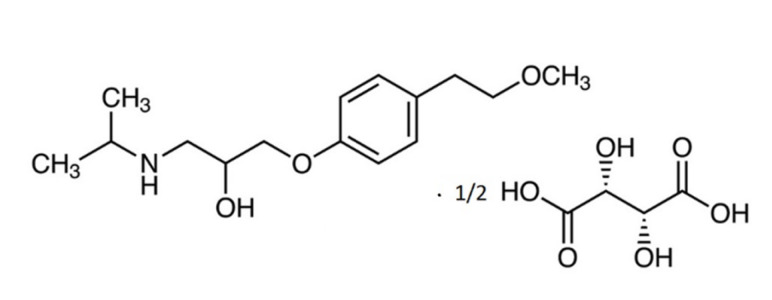
Formula of metoprolol tartrate (MT).

**Figure 2 ijms-23-13743-f002:**
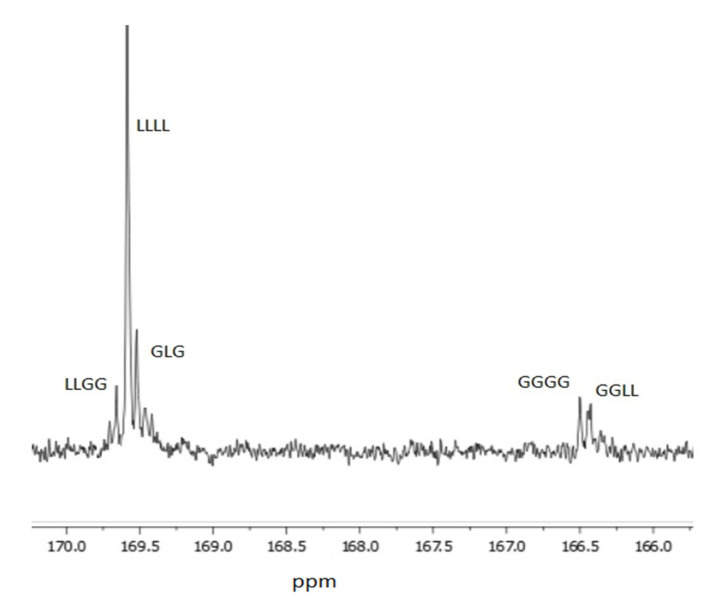
An approximation of the ^13^C NMR spectra of the PLGA carrier in DMSO-d_6_.

**Figure 3 ijms-23-13743-f003:**
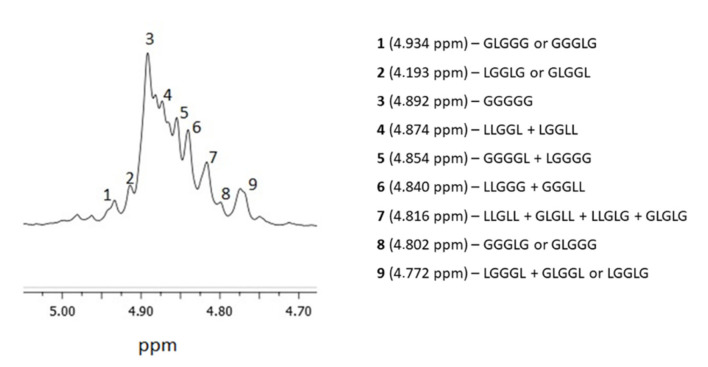
An approximation of the ^1^H NMR spectra of the PLGA carrier in CDCl_3_.

**Figure 4 ijms-23-13743-f004:**
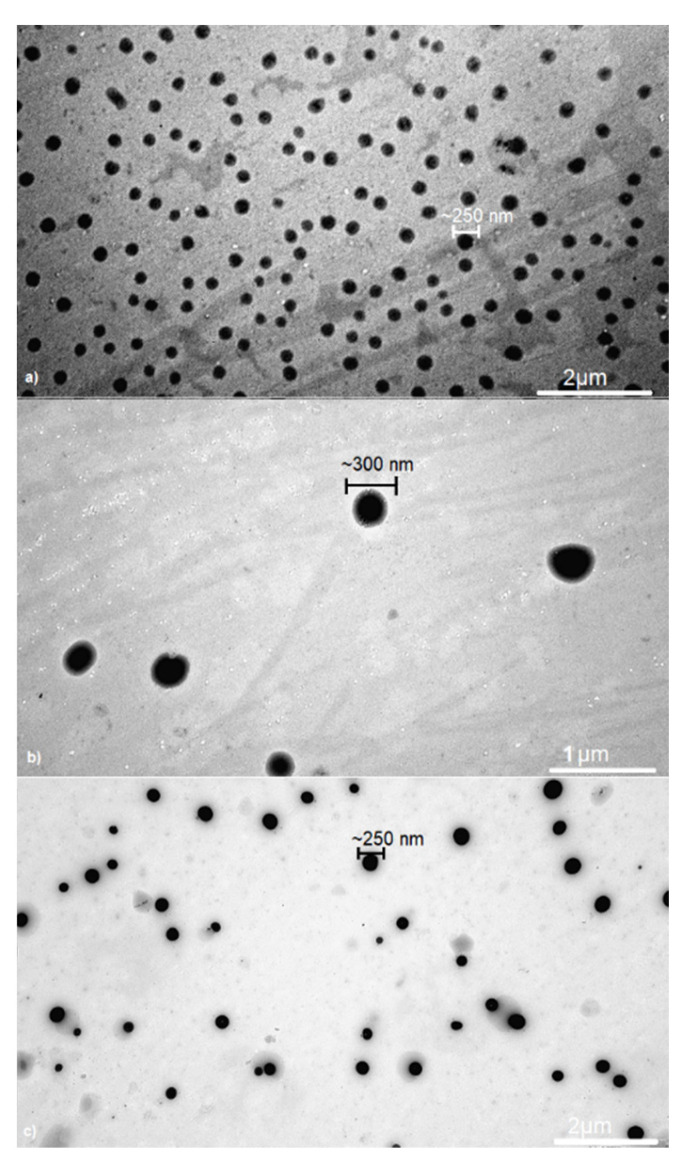
TEM image of: (**a**) O1 sample, (**b**) O2 sample, (**c**) O3 sample.

**Figure 5 ijms-23-13743-f005:**
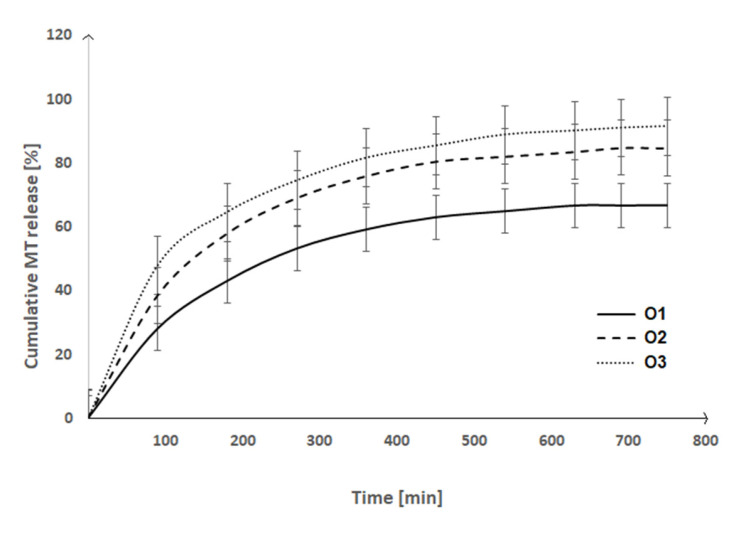
The MT release profile from the prepared biodegradable nanoparticle samples.

**Figure 6 ijms-23-13743-f006:**
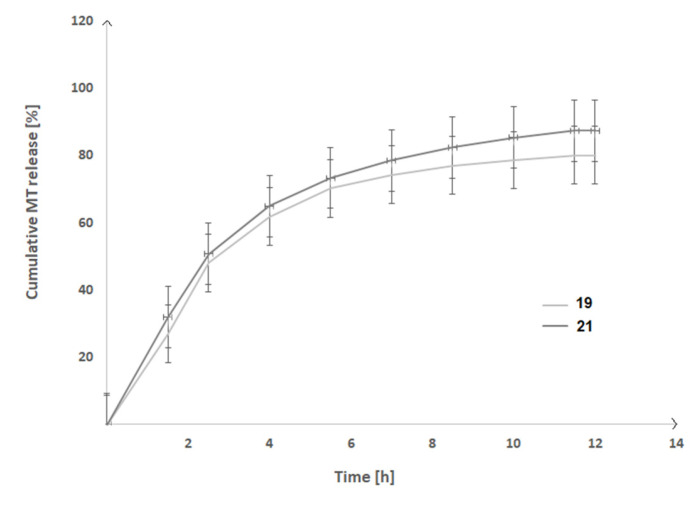
The MT release profile from the prepared thermosensitive LSs.

**Table 1 ijms-23-13743-t001:** Preparation and characterization of the biodegradable carrier.

Molar Ratio of PEG/GL/LLA	Molar Ratio of Zn/Monomers	Reaction Time [h]	Yield [%]	Temp. [°C]	Conv. LLA ^a^	Conv. GL ^b^	l_GG_ ^c^	l_LL_ ^d^	R ^e^	*M*_n_^f^ [g/mol]	*Đ* ^f^	T_II_ ^g^ [%]
1/10/1990	1/100	48	85	125	0.92	0.7	1.51	1.04	0.73	14700	1.45	12.3

where: ^a^ LLA conversion, ^b^ GL conversion, ^c^ the average lengths of glycolidyl blocks, ^d^ the average lengths of lactidyl blocks, ^e^ randomness degree, ^f^ determined by SEC-MALLS, ^g^ the second mode of transesterification.

**Table 2 ijms-23-13743-t002:** The results of the neutral red uptake test at the highest concentrations of tested extracts [1 mg/mL] in contrast to the untreated control.

Sample	Cells Viability ± SD
PLGA	107 ± 3
LTX *	2 ± 3
PE *	108 ± 3

* LTX (latex) and PE (poliethylene) were used as the reference materials (highly cytotoxic and with no cytotoxicity, respectively).

**Table 3 ijms-23-13743-t003:** The *umu*-test results for the highest concentrations of tested extracts [1 mg/mL].

	−S9 ^a^		+S9 ^b^	
**Sample**	**G ± SD**	**IR ± SD**	**G ± SD**	**IR ± SD**
PLGA	1.19 ± 0.02	0.63 ± 0.10	1.21 ± 0.06	0.63 ± 0.05
Solvent control	1.02 ± 0.03	0.92 ± 0.17	1.00 ± 0.05	0.89 ± 0.09
Positive control	0.74 ± 0.14	12.15 ± 1.31	0.77 ± 0.07	8.41 ± 0.95
Negative control	1.00 ± 0.06	1.00 ± 0.23	1.00 ± 0.06	1.00 ± 0.09

^a^ without metabolic activation; ^b^ with metabolic activation.

**Table 4 ijms-23-13743-t004:** Characterization of the developed nanoparticle samples loaded with MT.

Sample	MT [mg]	PLGA [mg]	Tween 80 Concentration [%]	Encapsulation Efficiency [%]	Size [nm]	Zeta Potential [mV]	PDI
O1	125	200	1.0	33	225	−18.95	0.14
O2	150	200	1.0	38	269	−18.53	0.03
O3	200	200	1.0	43	235	−20.95	0.01

**Table 5 ijms-23-13743-t005:** Data analysis of MT release from the prepared nanoparticles.

Sample	Zero-Order	First-Order	Higuchi Model	Korsmeyer-Peppas
O1	R^2^ = 0.812	R^2^ = 0.877 ^a^	R^2^ = 0.905	R^2^ = 0.953
			K_H_ = 14.804 ^b^	n = 0.399 ^c^
O2	R^2^ = 0.798	R^2^ = 0.915 ^a^	R^2^ = 0.894	R^2^ = 0.928
			K_H_ = 24.095 ^b^	n = 0.834 ^c^
O3	R^2^ = 0.840	R^2^ = 0.966 ^a^	R^2^ = 0.925	R^2^ = 0.956
			K_H_ = 32.303 ^b^	n = 0.301 ^c^

^a^ R^2^ values indicating zero-order or first-order kinetics of the MT release; ^b^ Release rate constant (Higuchi model); ^c^ The release exponent (Korsmeyer-Peppas model).

**Table 6 ijms-23-13743-t006:** Compositions of the prepared thermosensitive LSs loaded with MT based nanoparticles.

Sample	P188 [%]	P407 [%]	MT [%]	O3 [%]	HPC [%]	PVP [%]	Tween 80 [%]	Sodium Alginate [%]
1	15	15			1			
2	15	15				1		
3	15	15					3	
4	15	15						0.6
5	15	20			1			
6	15	20				1		
7	15	20					3	
8	15	20						0.6
9	15	15	0.5		1			
10	15	15	0.5			1		
11	15	15	0.5				3	
12	15	15	0.5					0.6
13	15	20	0.5		1			
14	15	20	0.5			1		
15	15	20	0.5				3	
16	15	20	0.5					0.6
17	15	15		1.75	1			
18	15	15		1.75		1		
19	15	15		1.75			3	
20	15	15		1.75				0.6
21	15	20		1.75	1			
22	15	20		1.75		1		
23	15	20		1.75			3	
24	15	20		1.75				0.6

**Table 7 ijms-23-13743-t007:** Gelation temperature values for the prepared thermosensitive LSs.

Samples	Gelation Temperature [^°^C]
1	39
2	45
3	36
4	39
5	38
6	41
7	37
8	38
9	37
10	44
11	40
12	35
13	36
14	40
15	39
16	38
17	36
18	42
19	36
20	35
21	35
22	40
23	38
24	38

**Table 8 ijms-23-13743-t008:** Gel strength, viscosity and gelation time parameters for the prepared thermosensitive LSs.

**Sample**	**Gel Strength [sec]**	**Viscosity [mPa·s]**	**Gelation Time [min]**
12	17	917.3	-
13	16	642.9	-
17	45	745.23	-
19	20	10229.9	3
20	13	1032.99	-
21	14	4225.71	4

**Table 9 ijms-23-13743-t009:** The results of the neutral red uptake test at the highest concentrations of the tested extracts in contrast to the untreated control.

Sample	Cells Viability ± SD [%]
19	82 ± 3
21	88 ± 1
LT	0 ± 0
PE	102 ± 1

LT—latex, reference cytotoxic material. PE—polyethylene foil, reference non-cytotoxic material.

**Table 10 ijms-23-13743-t010:** Data analysis of MT release from the developed thermosensitive LSs.

Sample	Zero-Order	First-Order	Higuchi Model	Korsmeyer-Peppas
Sample 19	R^2^ = 0.773	R^2^ = 0.893 ^a^	R^2^ = 0.870	R^2^ = 0.878
			K_H_ = 12.804 ^b^	n = 0.461 ^c^
Sample 21	R^2^ = 0.838	R^2^ = 0.977 ^a^	R^2^ = 0.920	R^2^ = 0.926
			K_H_ = 13.374 ^b^	n = 0.449 ^c^

^a^ R^2^ values indicating zero-order or first-order kinetics of the MT release; ^b^ Release rate constant (Higuchi model); ^c^ The release exponent (Korsmeyer-Peppas model).

## Data Availability

The data presented in this study are available on request from the corresponding author.

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
