# Peer review of "Development and Comprehensive Characteristics of Thermosensitive Liquid Suppositories of Metoprolol Based on Poly(lactide-co-glycolide) Nanoparticles"

_ijms, 2022, doi:10.3390/ijms232213743_

Round 1

Reviewer 1 Report

Thermosensitive liquid suppositories (LSs) carrying the model antihypertensive drug metoprolol tartrate (MT) were developed and evaluated. The nanoparticle system was based on a biodegradable copolymer synthesized by ring-opening polymerization (ROP) of glycolide (GL) and L,L-lactide (LLA). Biodegradable nanoparticles loaded with the model drug were produced by the o/o method at the first stage of the investigation. Depending on the concentration of the drug in the sample, from 66 to 91% of MT was released over 12 hours, according to first-order kinetics. Then, thermosensitive LSs with MT-loaded biodegradable nanoparticles were obtained by a cold method and their mechanical and rheological properties were evaluated. I think it is an interested work, but some problems should be solved before published in IJMS.

1. Why the molar ratio of GL/LLA is 10/90 in the biodegradable PLGA carrier? How about the other molar ratios of GL/LLA (20/80, 30/70, 40/60 …)?

2. In Table 5, the various thermosensitive LSs compositions are different, which one is the best important composition in thermosensitive LSs? And what is the influence law of this composition on thermosensitive LSs?

3. To evaluate the drug release profiles, the collected data were also fitted to mathematical models, especially the zero-order, first-order, Higuchi, and Korsmeyer-Peppas. You must add the fitted plots before Table 9.

4. In line 346, some references should be attached after ‘suggesting that the drug was primarily released via diffusion (n 0.45).’

5. Why you use different solvents for 1H NMR and 13C NMR?

6. Some errors must be corrected, such as lin 415 (Et)2Zn change to ZnEt2, line 451 13C NMR (CDCl3, 300 MHz, δH, ppm): ): change to 13C NMR (CDCl3, 300 MHz, δC, ppm):, no unified format in the References …

Author Response

Kindly please see the attachment for our responses.

Reviewer 2 Report

Bialik et al., present metoprolol (MT) loaded nanoparticles in a thermosensitive liquid suppository formulation for rental administration. The drug-loaded PEG-PLGA nanoparticles are encapsulated in poloxamer-based thermosensitive gel. It is an interesting study and readers would find it useful for designing rectal delivery systems.   The authors characterized the chemical and physical properties of the nanoparticles and gels in this article. While the authors presented in-depth analyses of drug-loaded nanoparticles, it seems that the gel analyses are not as thorough. If available, could the authors include some visual images of gels? Did the authors observe any effect of the mucoadhesive polymer on gelation temperature? How did the authors determine concentrations for P188 and P407, why 15%, not less or more than 20%? For example, for sample 17, if the concentrations for P188 and P407 remain at 15% for both and the HPC concentration doubles to 2%, how would this affect the gelation time and strength? Also, it would be helpful if the authors could explain in detail:  (i)how the gel strength is determined and the data should be interpreted. According to Table 7, gel strength is 20 sec for the gel with the highest viscosity while gel strength is 45 sec for the relatively low viscosity sample.  (ii)whether it correlates to viscosity or gelation time at all.    The authors should consider including analyses on the gel stability, disintegration, and degradation. For example, how long does it take to disintegrate/degrade at 37C? Are MT-nanoparticles evenly distributed within the LS when LS is gelled?

Author Response

(The authors gave the same response as above.)

Reviewer 3 Report

In this report, Okedszka and coworkers have developed and characterized temperature sensitive liquid suppositories (LS) that are loaded with a model antihypertensive drug ,metoprolol tartrate (MT) . These LS use biodegradable nanoparticles and the authors have optimized their rheological and mechanical properties for prospective rectal administration. The work is innovative and the use of such  novel, thermosensitive formulations may be an effective alternative to hypertension treatment, particularly for unconscious patients, or those  with mental illnesses, older patients,as well as  children. Overall the work is well done and reported in a clear manner that will be understood by propsoectuve readers of this journal.

Author Response

(The authors gave the same response as above.)

Round 2

Reviewer 1 Report

According to authors' responses and revisions, I think this paper could be published in IJMS. 

Author Response

Dear Reviewer,
We would like to express our appreciation for your kind assessment of our work.
Kind regards,
Authors

Reviewer 2 Report

Thank you for addressing the comments.

Author Response

(The authors gave the same response as above.)
